# The Effect of Craniofacial Manual Lymphatic Drainage after Moderate Traumatic Brain Injury

**DOI:** 10.3390/healthcare11101474

**Published:** 2023-05-18

**Authors:** Wilmer Danilo Esparza, Arian Ramón Aladro-Gonzalvo, Antonio Ruíz-Hontangas, Daniela Celi, María Belén Aguirre

**Affiliations:** 1Facultad de Enfermería, Pontificia Universidad Católica del Ecuador, Quito 170143, Ecuador; 2School of Physical Therapy, Universidad de Las Américas, Quito 170513, Ecuadorbethe7278@gmail.com (M.B.A.); 3Faculty of Health Sciences, Universidad Europea de Valencia, 46010 Valencia, Spain

**Keywords:** craniofacial manual lymphatic drainage, moderate traumatic brain injury, cerebral edema, cranial pain

## Abstract

Previous studies suggest that craniofacial manual lymphatic drainage (MLD) facilitates brain fluids clearance, reducing intracranial pressure and reabsorbing chronic subdural hematoma. This study aimed to explore the effect of craniofacial MLD in combination with pharmacological treatment for improving cranial pain intensity, vital signs, and cerebral edema (Hounsfield units, HUs) in moderate traumatic brain injury (mTBI). Patient 1 received pharmacological therapy, while patient 2 received both pharmacological and craniocervical MLD treatment. Patient 2 showed decreased cranial pain intensity and systolic blood pressure (66%–11.11%, respectively) after two 30 min daily sessions of treatment for three days. HUs in the caudate nucleus of both hemispheres (left 24.64%–right 28.72%) and in the left temporal cortical gray matter increased (17.8%). An increase in HU suggests a reduction in cerebral edema and vice versa. For patient 1, there were no changes in cranial pain intensity, but a slight increase in the systolic blood pressure was observed (0%–3.27%, respectively). HUs decreased in the temporal cortical (14.98%) and caudate nucleus gray matter (9.77%) of the left and right cerebral hemispheres (11.96%–16.74%, respectively). This case study suggests that craniofacial MLD combined with pharmacological treatment could reduce cerebral edema, decrease head pain intensity, and maintain vital signs in normal physiologic values in patients with mTBI.

## 1. Introduction

Traumatic brain injury (TBI) is an acute brain insult caused by an external mechanical force. The disruption affects neurons, glia, axons, and blood vessels and could involve cerebral edema (CE) [1]. The processes contributing to the development of CE are the interruption of the integrity of the blood–brain barrier, the regulation of cell volume via various ionic pumps, oncotic gradients, and inflammatory responses [2]. Cytotoxic cerebral edema affecting the gray matter is most often found after cerebral ischemia, typically secondary to TBI or stroke [3]. Experiments on animals and humans show the existence of a specialized cerebral clearing network [4] where fluid flows towards the venous perivascular and perineuronal spaces, draining waste products and excess fluid into meningeal lymphatic vessels and cervical lymph nodes (CLNs) [5]. This specialized clearing pathway mediated for astrocytes was named the glymphatic system [6]. The glymphatic system is fundamental for (i) brain nutrient delivery, particularly glucose supply; (ii) traffic and the distribution of apolipoprotein E isoforms produced in the choroid plexus; (iii) astrocytic paracrine signaling with lipid molecules; and (iv) extracellular metabolites and waste product drainage [7]. Recently, the fundamental role of lymphatic vessels in mouse and human bones in mediating hematopoietic and bone regeneration has been demonstrated [8].

Regarding extracerebral clearance, a main perineural egress site is along the olfactory nerve through the cribriform plate towards lymphatic vessels of the nasal mucosa [9,10]. From here, the cerebral spinal fluid is drained to the cervical lymph nodes [11,12], subsequently reaching the venous circulation. A recent systematic review shows the reduced clearance of the glymphatic system after TBI damage [13]. Cerebral lymphatic dysfunction can result in cerebrovascular diseases [14], subarachnoid hemorrhage [5], and multiple microinfarctions [15]. Moreover, the impairment of this system is associated with the formation of neurotoxic protein aggregates that are involved in the progression of neurodegenerative disease [16].

CE persistence is associated with increased mortality and poor functional outcomes. Furthermore, the effects of chronic mTBI (moderate TBI) could affect executive functions, self-awareness, and quality of life [17]. Therefore, the main goal of treatment is to reduce cerebral edema and minimize its impact on the brain. Medications such as corticosteroids and diuretics are used to reduce inflammation and decrease the amount of fluid in the brain [3]. Pain is also a common complication in patients with traumatic brain injury and can be caused by the injury itself or by the surgery required to treat the injury [3,18]. Pain can be intense and hinder the patient’s recovery. Analgesics such as opioids and non-opioids are used to treat pain. However, due to the risk of addiction and other side effects, it is important that these medications are administered under medical supervision and carefully tailored to the needs of the patient [19].

Manual lymphatic drainage (MLD) is a non-pharmacological and specialized treatment that consists of a gentle skin-stretching massage. This treatment of physical therapy improves the return of lymph/fluids to the circulatory system [20]. Specifically, MLD increases the amplitude and frequency of the contraction/relaxation of the lymphangions [21]. In addition, MLD has other effects, such as increasing venous flow, reducing fatigue, and raising the pain threshold [22]. Previous studies suggest that craniocervical MLD application is effective in reducing the intracranial pressure [23], promoting chronic subdural hematoma absorption [24]. However, little is known about the effect of craniocervical MLD on moderate TBI. Thus, this study aimed to explore the effect of MLD as an adjuvant therapy of the pharmacological treatment for improving cranial pain intensity, vital signs, and CE in mTBI. We support the hypothesis that, following a head injury, the MLD boosts the flow of cerebrospinal fluid, reducing cranial pain and not compromising the patient’s clinical status.

## 2. Materials and Methods

This was a case study of two male patients (patient 1, 59 years of age, and patient 2, 45 years of age) recruited at the hospital’s Neurosurgery Unit on the same day. To participate in the study, patients had to have suffered a moderate TBI (i.e., Glasgow Coma Scale = 9 to 12) and remain in the acute stage. They should not have presented: (1) loss of consciousness, (2) fractures (cervical or facial), (3) open wounds (cervical and/or facial), (4) pre-existing arterial hypertension, (5) underlying neurological conditions, (6) infections, or (7) cardiac decompensation.

Both patients had a left parietal fracture, but in patient 1, it was a slightly displaced double fracture. Both patients were right-handed and took the same drugs during the followed-up period: Paracetamol, 1 g, oral; Ketorolac, 30 mg, intravenous; Omeprazole, 40 mg, oral; Enoxaparin, 60 mg, intravenous; Lactulose, 25 mL, oral; and Tramal, 50 mg, intravenous. Patient 2 received craniocervical MLD, while patient 1 did not receive any physiotherapeutic treatment and was assessed at the same time as patient 2. The patients received no additional exercise or educational treatment. A total of six craniocervical MLD sessions using the “Leduc” method [9] were carried out twice a day for 3 days at the Neurosurgery Unit. The craniocervical treatment was performed by an experienced physiotherapist certified in MLD-type Leduc in the following order: (1) local lymph node drainage (posterior superficial cervical nodes, submental, submandibular, and preauricular nodes); (2) the clearance of maxillary, ethmoid, and frontal sinuses, as well as mental and zygomaticofacial foramen; and (3) the application of reabsorption, sweep, and effleurage maneuvers. Sinus clearance was also included due to the close anatomical relation between the meningeal lymphatic vessels, the paranasal sinuses, and CLNs [12]. The anterior superficial CLNs’ drainage was excluded from the treatment in order to avoid carotid sinus stimulation [25].

In both patients, cranial pain intensity, vital signs, and CE were assessed before the first session and after the final session of craniocervical MLD treatment. So as to assess cranial pain intensity (through the Visual Analogue Scale), patients were asked to judge the cranial pain intensity between 0 and 10. Vital signs (heart rate, “HR”; respiratory rate, “RR”; oxygen saturation, “SpO_2_”; blood pressure, “BP”; and body temperature, “BT”) were measured during the study to monitor the hemodynamic response of the patients and avoid clinical decompensation.

The CE was assessed with a computed tomography (CT) scanner (Siemens Syngo Somatom^®^, 2000, Frimley, UK) by a certified radiologist. The radiologist performed the assessment according to imaging techniques writing the HU values generated by the software and was blinded to the treatment. The images were recorded in Digital Imaging and Communication in Medicine Format. RadiAnt 2023.01 software was used to measure the gray matter density in Hounsfield Units (HUs) in the regions of interest (ROIs). The images were only measured in non-contrast brain CT. The measurements were performed in both brain hemispheres, in the cortical (e.g., temporal region, “TR”) and subcortical (e.g., caudate nucleus, “CN”) areas. In all images, an ROI of 4 mm^2^ was used. We assumed normal HUs for gray matter density around >40 [26]. The increase in HUs suggested a CE decrease [27].

### Statistical Analysis

The results are reported in the unit of measurement of each outcome variable, as well as the absolute and relative change score. A change score was considered clinically significant when a difference ±15% was observed.

## 3. Results

### 3.1. Cranial Pain and Vital Signs

The Glasgow Coma value was 11 for both patients. Cranial pain intensity decreased in patient 2 after craniocervical MLD treatment. In contrast, patient 1 did not have cranial pain intensity modifications. The vital signs registered did not show major changes, staying close to the normal ranges in both patients. Nevertheless, patient 2 showed a progressive decrease in the systolic BP, while patient 1 manifested an increase during the treatment period (Table 1).

### 3.2. Cerebral Edema

Gray matter density changes were observed in both subjects (Figure 1 and Figure 2). In patient 2, the HUs increased in the cortical gray matter of the left cerebral hemisphere and the caudate nucleus of both hemispheres. In contrast, the HUs decreased in the cortical (temporal region) gray matter of the right cerebral hemisphere. In patient 1, a decrease in the HUs was observed in all evaluated regions, indicating an increase in CE (Table 2).

## 4. Discussion

This case report showed a decrease in cranial pain intensity, systolic BP, and partial CE in the patient with moderate TBI who received craniocervical MLD. No adverse effect was observed during the treatment. Cranial pain intensity decreased by 66% in patient 2, while in patient 1, there was no modification at the end of treatment. No reports have been found regarding cranial pain reduction after craniocervical MLD in patients with an mTBI. The closest report to this anatomical area has been the effect achieved in increasing the pressure pain threshold of the trapeze muscle in healthy subjects after cervical MLD [28]. After a TBI, changes in brain fluid disturb the intracranial pressure, producing nerve cell injuries and cranial pain [29]. Thus, we assumed that the craniocervical MLD could normalize the excess fluid in the brain, with positive consequences on cranial pain reduction. Furthermore, the craniocervical MLD provides light stimulation to the skin and may also have a pain-inhibiting effect by stimulating the parasympathetic tone [30]. Future studies should consider an exhaustive pain assessment in patients with a TBI after MLD treatment, although the patient’s clinical condition may be a significant constraint for the assessment.

Regarding the vital signs, a decrease in systolic BP was observed in patient 2, while the other values oscillated within the normal limits in both patients. Patient 2 began with high blood pressure; this could have been due to the acute stress of the injury, and it may have decreased spontaneously even without specific intervention. However, a systolic BP decrease was previously reported in healthy subjects [28,31] and patients [23] after MLD. Systolic BP control/monitoring can be decisive since hypotension is an indicator of poor prognosis and a risk factor for increased mortality following TBI. A recent study found that systolic BP <100 mm Hg in patients ≤60 years old with severe TBI was associated with a significant increase in mortality [32]. In relation to SpO_2_, both patients were significantly hypoxic before the intervention. Hypoxia could be caused by the acute phase of the injury as well as the altitude. On the one hand, the development of CE following TBI is a complex heterogeneous process that may include other clinical consequences such as hypoxia, hypotension, hyperthermia, and seizures [2]. On the other hand, the patients were treated at 2800 m above sea level where the average oxygen saturation is 94% (±2%) [33].

Concerning the CE, it stands out that no patient had perihematomal edema, defined as a low-gray-matter-density area (HUs ranged from 5 to 33) [34]. The most relevant finding was an increase in the HUs in the temporal (in the left cerebral hemisphere) and caudate nucleus (in both cerebral hemispheres) regions of patient 2. This edema decrease is consistent with the study by Roth et al. [23], where an intracranial pressure reduction was observed in patients with severe brain disorders after craniocervical MLD. The decrease in the HUs in the cortical (temporal region) gray matter in the right cerebral hemisphere of patient 2 could have been due to: (1) venous system deficiency or obstruction and (2) venous vessel structure differences [35].

Other aspects that were observed were not the aim of the study included diuresis, visual function, consciousness level, and patient satisfaction. Patient 2 stated an increase in urine quantity as well as a less dark urine color. This finding could be important because a decrease in renal blood flow and the release of inflammatory substances may contribute to acute kidney injury in patients with TBI. To treat episodes of elevated intracranial pressure, osmolar agents such as mannitol and hypertonic saline are frequently administered in patients with TBI in intensive care units. Additionally, hemodynamic management based on the combined use of intravascular fluids and vasopressors is used. These agents have been associated with an increased risk of developing acute kidney injury [36]. MLD, due to its effect on the return of lymphs/fluids to the circulatory system [20], seems to contribute to improving renal function in these clinical circumstances. On the first day of treatment, patient 2 reported blurred vision, but on the third day of treatment, he noticed an improvement in his vision. A significant percentage of patients with mTBI report visual symptoms. Among the most common are disorders of extraocular movements and photophobia [37,38]. The physiotherapist also noticed that while patient 2 was drowsy during the first session, by the second day of treatment, he was more alert and engaged in more active communication. Research has highlighted that another consequence of TBI is increased sleepiness, especially in the early stages, caused by the damage of orexin/hypocretin neurons, whose activation involves wakefulness [39]. Finally, patient 2 expressed a pleasant feeling of relaxation after each session. Previous studies have reported an increase in relaxation after applying MLD for 15 min on the neck [40]. MLD also stimulates the autonomic nervous system, which would imply it reduces stress and blood pressure [41,42].

The limitations of the study are those of a case study, namely, the lack of generalizability and replicability, investigator bias, and difficulty establishing causality. However, we decided to conduct a case study because larger-sample research could be unethical, putting patients at unnecessary risk and limiting funds for other research. We also identified that the age difference between the subjects could have affected the normal recovery of clinical outcomes; in addition, patients with moderate TBI can have extremely variable recovery trajectories, and comparing one patient to another was very limited. In this sense, there is evidence showing that the gray matter volumetric in TBI presents diffusely between different areas of the brain [43]. Thus, it was not possible to consider one of the patients as a control subject. On the other hand, the use of a CT scan may have been a limitation, since Magnetic Resonance Imaging can offer a more sensitive measurement of the cerebral structures. Nevertheless, CT provides greater accessibility with faster image acquisition time, and this can be especially beneficial in acute settings with unstable patients [44].

As a strength, this study provides novel information about the use of physiotherapy in less-studied clinical conditions. Physiotherapy is crucial in the care of critically ill patients as it helps improve circulation, reduces the risk of complications, and decreases the intensive care unit stay time. Additionally, it promotes early mobility and rehabilitation, improving quality of life and reducing mortality in these patients. Our study also suggests that MLD could be a feasible and safe therapy, as well as identifying a new area of research that opens the door to randomized clinical trials with a larger sample.

## 5. Conclusions

This study suggests that two 30 min daily sessions during three days of craniocervical MLD decrease CE and the cranial pain intensity and maintain vital signs at normal physiologic values. The craniocervical MLD application in combination with pharmacological treatment may be a useful adjunctive therapy to stimulate fluid flow in patients with mTBI. Furthermore, this therapy seems to be feasible and safe.

## Figures and Tables

**Figure 1 healthcare-11-01474-f001:**
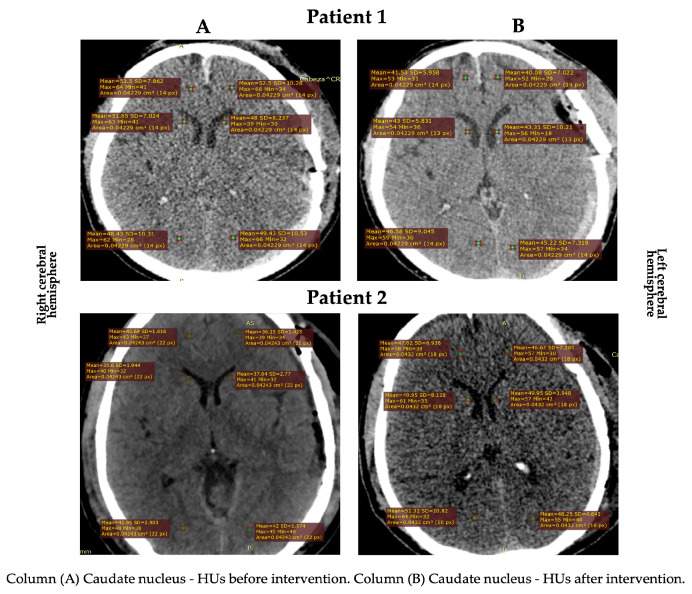
HU changes in the subcortical area in both patients.

**Figure 2 healthcare-11-01474-f002:**
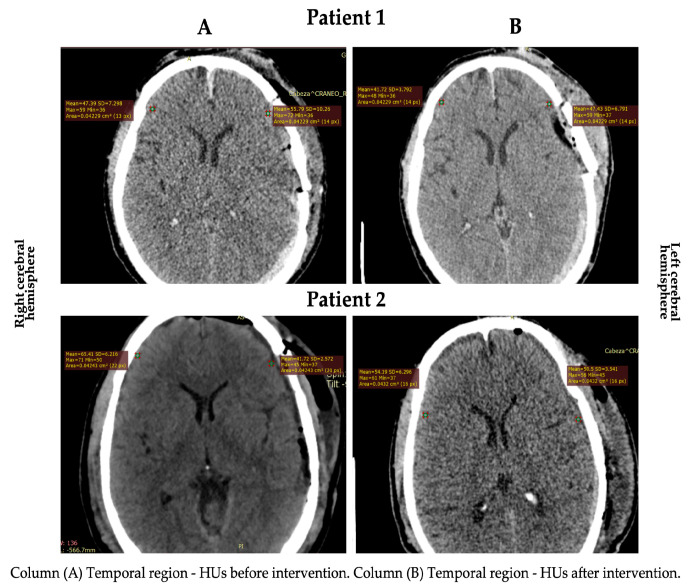
HU changes in the cortical area in both patients.

**Table 1 healthcare-11-01474-t001:** Changes in clinical and outcome scores.

	Before	After	Difference (%)
Cranial pain (cm)			
Patient 1	5	5	0 (=00.00)
Patient 2	6	2	4 (↓ 66.66)
Heart rate (lpm)			
Patient 1	88	75	13 (↓ 14.77)
Patient 2	59	60	1 (↑ 01.66)
Respiratory rate (rpm)			
Patient 1	14	15	1 (↑ 06.66)
Patient 2	13	17	4 (↑ 23.52)
Oxygen saturation SpO₂ (%)			
Patient 1	85	93	8 (↑ 08.60)
Patient 2	89	94	5 (↑ 07.44)
Systolic blood pressure (mmHg)			
Patient 1	118	122	4 (↑ 03.27)
Patient 2	144	128	16 (↓ 11.11)
Diastolic blood pressure (mmHg)			
Patient 1	70	66	4 (↓ 05.71)
Patient 2	66	80	14 (↑ 17.50)
Body temperature (°C)			
Patient 1	36.2	37	0.8 (↑ 02.16)
Patient 2	37.1	37.2	0.1 (↑ 00.26)

Note. The symbols ↓ (decrease) and ↑ (increase) represent de mean differences between pre and post-intervention with craniocervical MLD.

**Table 2 healthcare-11-01474-t002:** Values registered using computed tomography.

	Before	After	Difference (%)
RCH			
TR (HU)			
Patient 1	47.39	41.72	05.67 (↓ 11.96)
Patient 2	65.41	54.19	11.22 (↓ 17.15)
CN (HU)			
Patient 1	51.65	43.00	08.65 (↓ 16.74)
Patient 2	35.60	49.95	14.35 (↑ 28.72)
LCH			
TR (HU)			
Patient 1	55.79	47.43	08.36 (↓ 14.98)
Patient 2	41.72	50.50	08.78 (↑ 17.38)
CN (HU)			
Patient 1	48.00	43.31	04.90 (↓ 09.77)
Patient 2	37.64	49.95	12.31 (↑ 24.64)

RCH: right cerebral hemisphere; LCH: left cerebral hemisphere; TR: temporal region; HU: Hounsfield units; CN: caudate nucleus. The symbols ↓ (decrease) and ↑ (increase) represent de mean differences between pre and post-intervention with craniocervical MLD.

## Data Availability

All data generated and analyzed during this study are included in this manuscript.

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
