# Peer review of "The Effect of Craniofacial Manual Lymphatic Drainage after Moderate Traumatic Brain Injury"

_healthcare, 2023, doi:10.3390/healthcare11101474_

Round 1

Reviewer 1 Report

This is an important case report showing that the craniofacial MLD could facilitate the reduction of cerebral edema decrease head pain intensity, and maintain vital signs in typical physiologic values in patients with mTBI. The manuscript is well-written and presented.
A recent study shows the presence of lymphatic vessels in bones PMID: 36669473 . Please discuss this and its potential implications.

Author Response

[A] Dear reviewer, thank you very much for your valuable comments. We hope the changes introduced in the manuscript satisfy your requirements. We also apologize for the previous mistakes in the manuscript, and for the unclear parts of the document. We have included more information in the document to make the manuscript more understandable. We work hard with two main objectives, the first one is to provides novel information about the use of the physiotherapy in a less study clinical condition, as well as to promotes early recovery and improve quality of life in patients suffers with TBI. So, any comment that helps us to improve our purpose is always welcome.

Responses to Reviewer 1

[R1] This is an important case report showing that the craniofacial MLD could facilitate the reduction of cerebral edema decrease head pain intensity, and maintain vital signs in typical physiologic values in patients with mTBI. The manuscript is well-written and presented. A recent study shows the presence of lymphatic vessels in bones PMID: 36669473 . Please discuss this and its potential implications.

[A] Thank you for your suggestion. We believe that the findings of this study PMID: 36669473 decontextualize the discussion of our results; however, we believe that it is very relevant/novel information and we have included as follow (page 2 lines 80 to 81): 

“Recently, have been demonstrated the fundamental role of lymphatic vessels in mouse and human bones in mediating hematopoietic and bone regeneration”.

Reviewer 2 Report

Overall this is a fascinating topic that should be explored further in the literature. While your study is essentially a single case report, it does present some observations that might inform future investigators.

Title: This is fine.

Abstract: This needs to specify that Patient 2 received the MLD intervention whereas Patient 1 did not, otherwise the readers will initially erroneously assume that both subjects received the intervention, making the abstract’s results reporting very confusing.

There are also a couple of areas that need grammatical attention, including line 14 (I think this should read “facilitates the brain’s fluid clearance”) and line 22 (should be “increase in systolic blood pressure was [not ‘were’] observed).

Introduction: This is well written.

Methods:

Line 58. Range for moderate TBI is GCS 9-12, not just GCS 11. Is the intent to say that both of these subjects had GCS of 11. Some clarification in the wording is needed.

Lines 60-63: It looks like cranial fractures are one of the exclusion criteria, but earlier in the paragraph it is written that both of the subjects sustained parietal fractures, which are of course cranial fractures. It also seems to be stated that neurological conditions are excluded, which doesn’t make sense as a brain injury is a neurological condition. Is the intent to exclude individuals with underlying neurological conditions? If so, please insert the word “underlying” (“premorbid”).

Line 87: What does ROI stand for?

For the outcome measures, what standards are being used as a significant change in pain score? What is considered a significant change in Hounsfield Units? Basically, are the values presented in the results going to be considered probably significant or within measurement variation?

For the Hounsfield unit assessment, is this a simple number that comes up in the software, or is radiologist input or interpretation part of the process of obtaining this data point. If radiologist input/interpretation is involved, was the radiologist blinded to which patient received the MLD intervention and which did not? If radiologist manipulation has nothing to do with generating the Hounsfield Unit, but rather that it is purely a number automatically generated by the software, and therefore with minimal potential for interpretation bias, that should be stated.

Who the physiotherapist performing the intervention a certified lymphedema therapist? Were any other treatments provided during these treatment sessions that may have affected the outcome, such as exercise, education, etc?

Results:

In Table 1, it is striking that both of the subjects were significantly hypoxic in the “before” values, but this is not commented upon. This strikes me as atypical, and suggests that perhaps these are not typical moderate TBI patients. Are there clinical explanations for the hypoxia?

As mentioned above, it would be helpful to have some parameters around what is considered a significant threshold of change.

Discussion:

The biggest limitation is not stated here which is that this is a single case report. Individuals with moderate brain injury can have extremely variable recovery trajectories so the value of comparing one patient with one other patient highly limited. While all of the data is limited, the pain information strikes me as most compelling.  Regarding blood pressure, hard to say. Subject 2, who received the intervention, starts with high blood pressure, whereas subject 1, who did not receive the intervention, starts with normal blood pressure; subject 2’s blood pressure may have been from the acute stress of the injury and may have come down on it’s own even without the intervention. Saying it another way, subject 2’s blood pressure had more room to come down spontaneously, whereas subject 1’s did not. Also, was preexisting high blood pressure excluded? That is not listed as an exclusion. I can’t really comment on the value of the Hounsfield unit observations as I am not familiar with this measure and it is just hard to say without knowing cutoff values for significant change.

Author Response

[A] Dear reviewer, thank you very much for your valuable comments. We hope the changes introduced in the manuscript satisfy your requirements. We also apologize for the previous mistakes in the manuscript, and for the unclear parts of the document. We have included more information in the document to make the manuscript more understandable. We work hard with two main objectives, the first one is to provides novel information about the use of the physiotherapy in a less study clinical condition, as well as to promotes early recovery and improve quality of life in patients suffers with TBI. So, any comment that helps us to improve our purpose is always welcome.

Responses to Reviewer 2

Abstract:

[R2] This needs to specify that Patient 2 received the MLD intervention whereas Patient 1 did not, otherwise the readers will initially erroneously assume that both subjects received the intervention, making the abstract’s results reporting very confusing.

[A] Thank you for your suggestion. The modification was included as follow (page 1 lines 15 to 16): 

“Patient 1 received pharmacological therapy, while patient 2 received both pharmacological and craniocervical MLD treatment.”

[R2] There are also a couple of areas that need grammatical attention, including line 14 (I think this should read “facilitates the brain’s fluid clearance”) and line 22 (should be “increase in systolic blood pressure was [not ‘were’] observed).

The modifications were carried out.  (Line 12 and 21).

Methods:

[R2] Line 58. Range for moderate TBI is GCS 9-12, not just GCS 11. Is the intent to say that both of these subjects had GCS of 11. Some clarification in the wording is needed.

[A] Thank you for your suggestion. This information was included in Results section as follow (page 3 line 224):

“The Glasgow Coma value was 11 for both patients”

[R2] Lines 60-63: It looks like cranial fractures are one of the exclusion criteria, but earlier in the paragraph it is written that both of the subjects sustained parietal fractures, which are of course cranial fractures. It also seems to be stated that neurological conditions are excluded, which doesn’t make sense as a brain injury is a neurological condition. Is the intent to exclude individuals with underlying neurological conditions? If so, please insert the word “underlying” (“premorbid”).

[A] Thank you for your suggestion. This information was included as follow (page 2 lines 121-124):

“They should not present: 1) loss of consciousness, 2) fractures (cervical or facial), 3) open wounds (cervical and/or facial), 4) pre-existing arterial hypertension, 5) underlying neurological conditions, 6) infections, or 7) cardiac decompensation.”

[R2] Line 87: What does ROI stand for?

[A] Regions of interest (ROI) (page 3 line 217)

[R2] For the outcome measures, what standards are being used as a significant change in pain score? What is considered a significant change in Hounsfield Units? Basically, are the values presented in the results going to be considered probably significant or within measurement variation?

[A] Thank you for your questions. This information was included as follow (page 3 lines 225-227):

“The results are reported in the unit of measurement of each outcome variable, as well as the absolute and relative change score. A change score was considered clinically significant when a difference ± 15% was observed.”

[R2] For the Hounsfield unit assessment, is this a simple number that comes up in the software, or is radiologist input or interpretation part of the process of obtaining this data point. If radiologist input/interpretation is involved, was the radiologist blinded to which patient received the MLD intervention and which did not? If radiologist manipulation has nothing to do with generating the Hounsfield Unit, but rather that it is purely a number automatically generated by the software, and therefore with minimal potential for interpretation bias, that should be stated.

[A] Thank for your suggestion. This information was included as follow (page 3 lines 213-215):

“The radiologist performed the assessment according to imaging techniques writing the HUs value generated by a software and was blinded to the treatment”

[R2] Who the physiotherapist performing the intervention a certified lymphedema therapist?

[A] Thank for your question. This information was included as follow (page 2 line 133-134):

“The craniocervical treatment was performed by an experienced physiotherapist certified in MLD type Leduc..”

[R2] Were any other treatments provided during these treatment sessions that may have affected the outcome, such as exercise, education, etc?

[A] Thank for your question. This information was included as follow (page 2 line 130-131):

“The patients received no additional exercise or educational treatment.”

Results:

[R2] In Table 1, it is striking that both of the subjects were significantly hypoxic in the “before” values, but this is not commented upon. This strikes me as atypical, and suggests that perhaps these are not typical moderate TBI patients. Are there clinical explanations for the hypoxia?

[A] Thank for your question. This information was included as follow (page 6 lines 290-296):

“In relation to SpO2, both patients were significantly hypoxic before the intervention. Hypoxia could be caused by the acute phase of the injury as well as the altitude. On the one hand, the development of CE following TBI is a complex heterogeneous process that may include other clinical consequences such as hypoxia, hypotension, hyperthermia, and seizures [2]. On the other hand, the patients were treated at 2,800 meters above sea level where the average oxygen saturation is 94% (± 2%) [33].”

[R2] As mentioned above, it would be helpful to have some parameters around what is considered a significant threshold of change.

[A] Thank you for your questions. This information was included as follow (page 3 lines 225-227):

“The results are reported in the unit of measurement of each outcome variable, as well as the absolute and relative change score. A change score was considered clinically significant when a difference ± 15% was observed.”

Discussion:

[R2] The biggest limitation is not stated here which is that this is a single case report. Individuals with moderate brain injury can have extremely variable recovery trajectories so the value of comparing one patient with one other patient highly limited. While all of the data is limited, the pain information strikes me as most compelling. 

[A] Thank you for your suggestion. This limitation was included as follow (page 7 lines 349-355):

“We also identify that the age differences between subjects could conditioning the normal recovery of clinical outcomes, as well patients with moderate TBI can have extremely variable recovery trajectories, and comparing one patient to another was very limited. In this sense, there is evidence showing that the grey matter volumetric in TBI presents diffusely between different areas of the brain [43]. Thus, it was not possible to consider one of the patients as a control subject.”

[R2] Regarding blood pressure, hard to say. Subject 2, who received the intervention, starts with high blood pressure, whereas subject 1, who did not receive the intervention, starts with normal blood pressure; subject 2’s blood pressure may have been from the acute stress of the injury and may have come down on it’s own even without the intervention. Saying it another way, subject 2’s blood pressure had more room to come down spontaneously, whereas subject 1’s did not.

[A] Thank you for your suggestion. This information was included as follow (page 6 lines 283-286):

“The patient 2 began with high blood pressure, this could be due to the acute stress of the injury, and it may have decreased spontaneously even without a specific intervention. However, the systolic BP decrease was previously reported in healthy subjects [28,31] and patients [23] after MLD.”

[R2] Also, was preexisting high blood pressure excluded? That is not listed as an exclusion.

[A] Thank you for your question. This information was included as follow (page 4 line 122):

“…. 4) pre-existing arterial hypertension, 5) underlying neurological conditions, 6) infections, or 7) cardiac decompensation.”

[R2] I can’t really comment on the value of the Hounsfield unit observations as I am not familiar with this measure and it is just hard to say without knowing cutoff values for significant change.

[A] Thank you very much for your comment. We have included more information in the document to make the Hounsfield values more understandable. We also want to comment here that the degree of edema per unit volume can be measured by the water content in a unit volume. Different water contents present different HU on computed tomography (CT), and the appearance and disappearance could be manifested by the change of HU in local brain tissue.

Reviewer 3 Report

Dear Authors,

I feel obligated to indicate weak points of the manuscript:

1. this is rather case study

2. statistical analysis is poor

3. inclusion of two cases is insufficient for drawing the general conclusions

You should try to colect more data a prepare new manuscript. 

Author Response

[A] Dear reviewer, thank you very much for your valuable comments. We hope the changes introduced in the manuscript satisfy your requirements. We also apologize for the previous mistakes in the manuscript, and for the unclear parts of the document. We have included more information in the document to make the manuscript more understandable. We work hard with two main objectives, the first one is to provides novel information about the use of the physiotherapy in a less study clinical condition, as well as to promotes early recovery and improve quality of life in patients suffers with TBI. So, any comment that helps us to improve our purpose is always welcome.

Responses to Reviewer 3

[R3] statistical analysis is poor

[A] Thank you very much for your comment. We have included more information as follow (page 3 lines 225-227):

“The results are reported in the unit of measurement of each outcome variable, as well as the absolute and relative change score. A change score was considered clinically significant when a difference ± 15% was observed.”

[R3] inclusion of two cases is insufficient for drawing the general conclusions

[A] Thank you very much for your comment. We have included the limitations of the study highlighting the lack of generalizability of our results (page 7 line 346). Also have included the strengths as identifying a new area of research that open the door to randomized clinical trials with larger sample (page 7 lines 363-365).

[R3] You should try to colect more data a prepare new manuscript.

[A] Thank you very much for your comment. We have included more information in the document in order to improve its relevancy and strength (page 6 lines 306-328).

Round 2

Reviewer 2 Report

The paper is much improved. Great topic. Limited conclusions can be drawn since essentially a case report, but the paper supports that this modality should be investigated further.  A good English grammatical review is needed as the paper still has numerous minor grammatical errors.

Author Response

Responses to Reviewer 2

[R2] A good English grammatical review is needed as the paper still has numerous minor grammatical errors.

[A] Dear reviewer, thank you very much for your valuable suggestion. We also apologize for the previous language mistakes in the manuscript, and for the unclear parts of the document. We have corrected the grammatical errors to make the manuscript more understandable. We hope the changes introduced in the manuscript satisfy your requirements.
